# Diagnostic Utility of Temporal Muscle Thickness as a Monitoring Tool for Muscle Wasting in Neurocritical Care

**DOI:** 10.3390/nu14214498

**Published:** 2022-10-26

**Authors:** Andreas Maskos, Moritz L. Schmidbauer, Stefan Kunst, Raphael Rehms, Timon Putz, Sebastian Römer, Vassilena Iankova, Konstantinos Dimitriadis

**Affiliations:** 1Department of Neurology, University Hospital, LMU Munich, 81377 Munich, Germany; 2Institute for Medical Information Processing, Biometry, and Epidemiology (IBE), LMU Munich, 81377 Munich, Germany; 3Department of Neurology, Friedrich-Baur-Institute, University Hospital, LMU Munich, 81377 Munich, Germany; 4Institute for Stroke and Dementia Research (ISD), LMU Munich, 81377 Munich, Germany

**Keywords:** critical care, ultrasonography, temporal muscle, quadriceps muscle, muscular atrophy, sarcopenia

## Abstract

Temporalis muscle (TM) atrophy has emerged as a potential biomarker for muscle wasting. However, its diagnostic utility as a monitoring tool in intensive care remains uncertain. Hence, the objective of this study was to evaluate the diagnostic value of sequential ultrasound- and computed tomography (CT)-based measurements of TM thickness (TMT). With a prospective observational design, we included 40 patients without preexisting sarcopenia admitted to a neurointensive care unit. TMT measurements, performed upon admission and serially every 3–4 days, were correlated with rectus femoris muscle thickness (RFT) ultrasound measurements. Interrater reliability was assessed by Bland Altmann plots and intraclass correlation coefficient (ICC). Analysis of variance was performed in subgroups to evaluate differences in the standard error of measurement (SEM). RFT decline was paralleled by ultrasound- as well as CT-based TMT measurements (TMT to RFT: r = 0.746, *p* < 0.001; CT-based TMT to ultrasound-based RFT: r = 0.609, *p* < 0.001). ICC was 0.80 [95% CI 0.74, 0.84] for ultrasound-based assessment and 0.90 [95% CI 0.88, 0.92] for CT-based TMT measurements. Analysis of variance for BMI, Heckmatt score, fluid balance, and agitation showed no evidence of measurement errors in these subgroups. This study demonstrates the clinical feasibility and utility of ultrasound- and CT-based TMT measurements for the assessment of muscle wasting.

## 1. Introduction

Intensive care unit acquired weakness (ICU-AW), describing the neuromuscular dysfunction secondary to critical illness, is estimated to have an incidence of around 40% in patients receiving critical care [1,2]. The main contributors seem to be sepsis, multi-organ failure, underfeeding, and lack of mobilization [3]. Besides, the paucity of suitable biomarkers hinders the monitoring of disease and thereby complicates the assessment of therapeutic approaches with regard to clinical efficacy [4].

Methods for diagnosing ICU-AW include functional testing (e.g., Medical Research Council Scale (MRC) sum score, body mass index adjusted handgrip strength, walking speed), imaging studies, electrophysiology, and biopsy. However, most tests have their limitations in critically ill and sedated patients. In neurointensive care patients, higher amounts of sedatives and the high prevalence of co-existing neurological deficits make the assessment particularly challenging.

Therefore, a lot of attention is being paid to bedside methods such as ultrasound (US) as well as to strategies to extract information on muscle wasting (sarcopenia) from imaging performed in clinical routine to minimize potential hazards associated with the assessment for ICU-AW [5]. In a recent review on peripheral muscle ultrasound in the ICU, the authors depicted the importance of monitoring muscle wasting during the course of intensive care unit stays to assess and counteract the development of ICU-AW [6]. As a result, most ultrasound studies use rectus femoris thickness (RFT) to determine muscular atrophy as a surrogate marker of evolving sarcopenia and consequently ICU-AW [6]. As previously demonstrated, ICU-AW per se is also highly relevant to a number of ICU-related outcomes [7]. Feasible and accurate diagnostic tools are therefore necessary to monitor the development of ICU-AW. Recently, researchers have also started using measurements of temporalis muscle thickness (TMT) as a biomarker in non-ICU patients as an outcome predictor [8,9]. Estimation of temporalis muscle volume has been used as a sequential parameter for muscle wasting and was shown to correlate to early nutritional inadequacy in SAH patients in a retrospective study [10]. So far, only two studies used serial ultrasound measurements to determine changes in TMT. Hasegawa et al. introduced longitudinal temporal muscle ultrasound measurements in bedridden elderly individuals, whereas Anand et al. assessed its use in sepsis patients [11,12]. Hasegawa and colleagues could demonstrate that temporal muscle thickness decline correlated significantly with inadequate energy intake in bedridden older adults [11]. In contrast, Anand et al. questioned the usefulness of temporalis muscle as a monitoring tool for muscle wasting in sepsis patients [12].

It thus remains unclear whether TMT is of diagnostic utility to assess muscle wasting, nutrition status, and the development of ICU-AW of patients admitted to an intensive care unit in a longitudinal fashion, more so in neurocritical care patients. As sequential magnetic resonance imaging (MRI) is not feasible in critically ill patients, there is also a clinical need to determine the optimal diagnostic modality for TMT measurements.

The aim of this study was first to evaluate the utility of TMT compared to RFT in monitoring muscle wasting during the stay in a neurointensive care unit (NICU). Second, to investigate potential confounders influencing the precision of measurements in neurocritical care patients, and third, to compare US and CT-based measurements in their interrater reliability.

## 2. Materials and Methods

### 2.1. Screening and Eligibility

For this single-center prospective observational study, all consecutive patients admitted to the neurological intensive care unit were included if the age was above 18, the reason for ICU admission was of neurologic etiology, and if the expected length of stay was more than 7 days. Exclusion criteria were bifrontal craniectomy, confinement to bed prior to ICU admission, recent hospitalization, renal replacement therapy, other primary neuromuscular disease, disseminated cancer, and pregnancy. The protocol was approved by the local ethics committee of the Ludwig Maximilian University of Munich (Nr. 20-0644) for recruitment between April 2021 to April 2022.

Baseline data collected within 24 h after admission and over the course of ICU stay included age, sex, admission diagnosis, length of stay in ICU, days under mechanical ventilation, weight, height, body mass index (BMI), Glasgow Coma Scale Score (GCS), Confusion Assessment Method (CAM)-ICU-7 delirium severity scale, Richmond Agitation–Sedation Scale (RASS), Sepsis-Related Organ Failure Assessment score (SOFA), Acute Physiology and Chronic Health Evaluation (APACHE II) score and Nutrition Risk in Critically ill score (NUTRIC). Ultrasound, weighing, and laboratory tests (24 h-urine and blood parameters) were repeatedly conducted on days 3, 7, 10, 14, 17, and day 20 (or prior to discharge from ICU). For feasibility reasons, time spans of 12 h in either direction upon exact measurement day were accepted for image acquisition. The patient’s weight was obtained using a Radwag C315 weighing system. Body weight estimation was conducted as described by Freitag et al. [13] and nitrogen balance was calculated as published by Kim et al. [14].

### 2.2. Clinical Management

Patients received caloric and protein prescriptions according to the European Society of Parenteral and Enteral Nutrition (ESPEN) guidelines [15]. General targets were 25 kcal/kg/day and 1.5 g/kg/day according to the obtained bodyweight, which was adjusted after every bodyweight estimation in 3-to-4-day intervals. Values for mean achieved calories and mean achieved protein were calculated between measurement days.

### 2.3. Imaging Protocols

Esaote MyLabOmega with a 20 MHz linear probe was used to obtain images. Image acquisition of subcutaneous adipose tissue (SAT), rectus femoris thickness (RFT), rectus femoris cross-sectional area (RF CSA), and vastus intermedius thickness (VIT) was performed following the protocol provided by Galindo et al. on both lower extremities [16]. As suggested by Pardo et al., the probe was placed in the lower third of the connecting line between the anterior superior iliac spine and the upper border of the patella [17]. To ensure the same measurement location over the length of stay, we drew a line on the patient’s leg for probe placing. Device settings were kept identical for every patient and throughout the study period. In contrast to Chang et al., who moved the transducer cranially and parallel to the zygomatic arch until the temporal muscle was visible, we performed TMT measurements by placing the transducer on a fictive line between the zygomatic process of the frontal bone and the ipsilateral helix of the auricle (Figure 1), since this point can be identified more clearly in follow-up ultrasound acquisition processes and is furthermore comparable to the measurement point of TMT in CT [18,19]. TMT was defined as the distance between the temporal fascia and the lowest point of the temporal fossa (Figure 1). The transducer was placed rectangular to the skin with minimal pressure and bilateral measurements were performed if applicable. To reduce measurement interference, we instructed patients not to clench if not under sedation. To determine interrater reliability in ultrasound-based measurements, 20 patients were measured in parallel by two experienced examiners. The two observers were unaware of each other’s results and performed measurements independently.

Image review of CT scans was performed on a picture archiving and communication system (PACS) workstation. Determination of TMT on CT images was conducted as described by Katsuki et al. [19]. To assess interrater reliability for CT-based TMT measurements in all included patients, two observers performed these measurements independently and were unaware of each other’s results. Additionally, the observers for CT-based TMT measurements were not involved in the ultrasound measurement processes.

### 2.4. Sample Size and Statistical Analysis

After analyzing statistical power for the correlation of ultrasound-based TMT to RFT measurements in a cohort of 10 patients, the data indicated that at least 20 patients would be required for an α level of 0.05 and a β level of 0.20 (meaning a power of 0.80) with a correlation coefficient of r > 0.6. Since the assumptions of the sample size planning were based on limited pre-existing data on TMT, we decided to include 40 patients. For statistical analysis, the statistical computing language R version 4.0.4 including its packages ggplot2 and icc were used [20]. Variables are presented as frequencies with percentages, mean and standard deviation (SD) or as median and 25%/75% quartile. The relative changes of muscle diameter (Δd) at timepoint t (t) were calculated using the formula: Δd_t_ = (d_t_ − d_0_)/d_0_. In general, a two-sided *p* value ≤ 0.05 was considered statistically significant. To assess muscular atrophy, a Wilcoxon signed-rank test for paired samples referencing day 0 was used. Since the data do not suggest non-linear dependencies, the relationship among measured variables was investigated using Pearson correlation. As a significance test for correlation, Fisher’s Z transform was used. To account for potential hemiparesis in NICU patients, correlations were calculated for each side. Evaluation of inter-rater reliability (IRR) was determined by Bland Altmann plots and intraclass correlation coefficient (ICC) with the according to 95% confidence intervals (CI). For ICC, the following values were used to indicate levels of reliability: <0.50, poor reliability; 0.50–0.75, moderate reliability; 0.75–0.90, good reliability; >0.90, excellent reliability [21]. Analysis of variance for the standard error of measurement (SEM) was performed using BMI, RASS, Heckmatt score, and fluid retention normalized to body weight as group-defining variables [22]. After the first visualization of the SEM for TMT measurements for all included patients on day 10, clear outliers were excluded from dichotomization. As seen for the correlation test, observations were separated for each side to minimize the influence of hemiparesis. The dichotomization of the study cohort was then performed in the four groups defining variables mentioned above to analyze the effects of possible confounders on the SEM. The following aspects were considered: (i) analysis of effects of body weight with cut-off values for group one with a BMI < 30 versus group 2 with BMI values of 30 or above (ii) analysis of the muscle echogenicity influence on SEM with cut off values for Heckmatt score < 3 for group 1 versus 3–4 for group 2 (iii) analysis of effects of agitation with dichotomization for RASS < 0 versus RASS 0–4 (iv) analysis of the implication of fluid balance. Here, dichotomization of patients was performed according to fluid retention in the first ten days of intensive care stay, with group one having a fluid retention of less than 7.5% per kilogram bodyweight versus group 2 with more than 7.5% per kilogram body weight. To test statistical differences in the SEM between the subgroups, we used the F-test.

## 3. Results

### 3.1. Study Demographics and Nutritional Characteristics

Baseline data are summarized in Table 1. Overall, 40 patients were enrolled with a minimum ICU follow-up of 10 days. Twenty-six study participants stayed until measurement day 14, 21 until day 17 and 12 patients were monitored up to day 20. The median age was 60 years (IQR 52.75–73.5) and most patients were male (31/40). On admission, the mean BMI was 29.3 (SD 4.3) and the most frequent cause for ICU admission was cerebrovascular disease (32/40). The median length of ICU stay (LOS) was 20 days (IQR 13–34), with a median time of mechanical ventilation of 11.5 days (IQR 8.5–22.25). From day 1 to day 10, patients received a mean of 983.05 kcal/day (SD 310.12 kcal/day). A mean protein intake of 0.66 g/kg/day (SD 0.42 g/kg/day) was achieved in the same time span. Nitrogen balance as a surrogate marker for protein catabolism was negative on day 10 with a mean value of −8.5 g/day (SD 7.1 g/day).

### 3.2. Muscular Atrophy Characteristics

In multimodal measurements of TMT (CT and US) and in the ultrasound-based measurement of RFT, a substantial decline in muscle mass was detected during the ICU stay as presented in Figure 2. In ultrasound-based measurements, TMT declined from 10.9 mm (SD 1.6 mm) at baseline to 10.1 mm (SD 1.4 mm) on day 10 and 9.6 mm (SD 1.3 mm) on day 20, whereas RFT declined from 10.1 mm (SD 2.6 mm) at baseline to 8.5 mm (SD 1.9 mm) and 7.6 mm (SD 1.4 mm) at day 10 and day 20, respectively. TMT measured in serial CT scans declined from 9.0 mm (SD 2.2 mm) at baseline to 7.6 mm (SD 1.2 mm) on day 10 and 6.3 mm (SD 0.7 mm) on day 20.

### 3.3. Inter-Rater Agreement

Figure 3 shows the Bland–Altmann plots of TMT measurements visualizing inter-rater agreement between examiner 1 (E1) and examiner 2 (E2). The mean bias was −0.189 mm, and the upper limit of agreement (LOA) was 1.55 mm, while a lower LOA of −1.93 mm was acknowledged in ultrasound-based TMT measurements. As for ICC in ultrasound-based measurements, we observed a value of 0.797 (95% CI 0.737, 0.844). Furthermore, we analyzed 156 CT images of 39 patients retrospectively with two blinded observers, where the mean bias was −0.005 mm with an upper LOA of 2.03 mm and a lower LOA of −2.04 mm. Moreover, the ICC of E1 and E2 showed overall good agreement with a value of 0.899 (95% CI 0.88, 0.92).

### 3.4. Correlation of TMT Muscle Loss between Ultrasound and CT Measurement

Ultrasound-based RFT and ultrasound-based TMT (left side r = 0.746, *p* < 0.001; right side r = 0.631, *p* < 0.001) as well as ultrasound-based RFT and CT-based TMT measurements (left side r = 0.609, *p* < 0.001; right side r = 0.592, *p* < 0.001) showed significant correlation. (Figure 4).

### 3.5. Analysis of Variance for Possible Confounders

BMI, Heckmatt score, RASS, and fluid retention did not significantly influence the standard error of measurement in our analysis. Graphic visualization of the different aspects can be seen in Figure 5.

## 4. Discussion

In this single-center observational study, ultrasound- and CT-based TMT measurements exhibited diagnostic utility to monitor muscle wasting in a cohort of neurocritical care patients as (i) TMT and RFT showed comparable results in sequential measurements and (ii) high inter-rater reliability in both CT- and ultrasound-based measurements up to 20 days was noticed and (iii) robustness to potential confounders with high prevalence in neurocritical care such as BMI, a difference of muscle echogenicity, agitation and changes in fluid balance was observed.

When establishing an imaging protocol as a diagnostic strategy to monitor muscle wasting as an important aspect of ICU-AW in neurocritical care, important key points are the selection of the most suitable muscle, the most pragmatic and accurate method of measurement, and the validity of sequential measurements with analysis of potential confounders.

Concerning the most appropriate muscle, we could demonstrate that TMT is a promising alternative to the well-established RFT. Average baseline values in studies assessing TMT via ultrasound range from 4.9 mm to 9.7 mm depending on the measuring point [18,23]. Both TMT and RFT decline in our study demonstrated a similar extent and pattern compared to previous studies, thereby highlighting external validity for TMT measurement [10,11,24,25]. Moreover, the decline in thickness correlated significantly between both muscles.

In previous critical care trials on the topic of muscle wasting and ICU-acquired weakness, the most frequently examined sites included the tibialis anterior, rectus femoris, and vastus intermedius muscle [6]. However, RFT measurement bears some disadvantageous aspects concerning confounders and clinical relevance, especially in neurocritical care patients. First, the influence of paralysis on thickness decline is not known. Second, for repeated measurements, patient positioning requires to be completely flat, with legs in a relaxed horizontal position to reach optimal inter-rater and intra-rater reliability [17]. As rigorous positioning regimes with suboptimal leg positioning for ultrasound assessment of RFT are part of neurocritical care in patients with critical intracranial pressure, RFT seems to be less suitable.

Moreover, studies showed an earlier and more severe atrophy in muscles with type 2 fibers being dominant in thigh muscles, but underrepresented in jaw-closing muscles, especially in temporalis muscle [24,26,27]. This might partly explain the higher reduction of RFT compared to TMT in our data. Given these differences, the clinical significance of the respective muscles is of particular interest.

Regarding the most suitable method, ultrasound for TMT evaluation was used by different working groups in different methodical setups and patient cohorts. Studies with single-time measurements focused mainly on capturing sarcopenia as a prognostic marker, on the reliability of measurements, and on the influences of different postures and voluntary muscle contraction on ultrasound-guided imaging [18,23]. Similar to previous studies, the ultrasound-based measurement of TMT in our study was feasible and showed a good inter-rater reliability [18,23]. Longitudinal measurements introduced by Hasegawa et al. were carried out focusing on the correlation of energy intake inadequacy to temporal muscle atrophy in bedridden elderly non-ICU patients [11]. Only one further study intended to monitor sarcopenia in an ICU setting by performing serial ultrasound measurements of TMT in sepsis patients [12]. Here similar atrophy characteristics, but no significant correlations between RFT and TMT were acknowledged. This disagreement with our data might be explained by differences in patient selection (only patients with sepsis and not accounted for possible preexisting sarcopenia) as well as by the image acquisition processes and length of observation.

Although TMT appears to be appropriate for assessing muscle wasting and there is some evidence suggesting TMT correlates with dysphagia as well as hand grip strength and therefore possibly also with ICU-AW [28,29]. Larger prospective studies are required to answer the question of whether TMT is a suitable surrogate to evaluate dysphagia or whether it can drive weaning and nutritional strategies. However, the following points appear to make ultrasound-based TMT measurements attractive for monitoring muscular atrophy, particularly in neurocritical care. First, the measurement point is easy to identify using the anatomical landmarks described above. Second, measurements can be achieved in a ‘one stop shop’ approach by combining it with ultrasound examinations frequently applied in neurocritical care (optic nerve sheath diameter (ONSD), third ventricle diameter, transcranial doppler (TCD)) [30]. Third, and inherent to ultrasound techniques in general, it allows bedside, cost-effective and real-time assessment. Fourth, as no effects of possible confounders could be detected, ultrasound-based TMT measurements seem to be a reliable method.

CT- and MRI-based TMT measurements based on routine imaging scans have shown their practicality in evaluating preexisting sarcopenia and patient outcome [31]. Steindl et al. even defined standard values of TMT in MRI in the Caucasian population [29]. By using semi-automated software for CT-based TMT measurements, Onodera et al. could demonstrate the detection of developing sarcopenia in SAH patients [10]. Paralleling this work, our study presented a similar progressive TMT decline in CT-based measurements. This might be of relevance since brain imaging methods are widely used in NICU patients. Yet, not all patients need serial CT scans and timing of imaging, with the vast majority of scans at the early phase of the disease, might hinder detailed monitoring of long-term outcomes such as sarcopenia or ICU-AW.

As shown in our study the decline pattern of TMT matches the one of RFT, and both methods (Ultrasound and routine CT scans) seem to adequately capture muscle wasting. However, the value of sequential measurements must be assessed in prospective interventional studies with different nutritional and mobilization strategies based on TMT decline in preset time points. The fusion of different imaging processes could possibly improve measurement accuracy and should be looked at in future studies.

This is the first trial that systematically evaluates the relevance of different methods and muscles with short-interval follow-up measurements in NICU patients. Due to its prospective nature, the rigorous inclusion criteria, and the monitoring of clinical, nutritional, and laboratory data, the study also allows an evaluation of relevant confounders.

This study had several limitations. As this was a single-center study investigating NICU patients with a prolonged stay in NICU, caution must be taken in generalizing the findings. Nevertheless, the atrophy of rectus femoris in our study was similar to findings in more general populations with less strict inclusion criteria, which supports the external validity argument. Moreover, based on the cut-off points of Steindl and colleagues only three recruited study patients in this study showed TMT values below two standard deviations, meaning that only 7.5% were defined as pre-sarcopenic, which coincides with the low rate of pMRS score greater than 0 in our study cohort and speaks for good internal validity. Also, as we used a slightly different angle for ultrasound-based TMT measurements, comparability with other studies is hindered. Yet, studies using volumetric methods (which are independent of the exact measuring point for TMT) seem to have similar results. Finally, the pattern of decline was in line with the one found in previous studies, which alleviates the need for a precise measurement point for single-center studies. However, to improve comparability across studies a consensus statement must be made concerning the exact measuring method and point [32]. There were some missing data for CT-based TMT due to surgery or trauma or the sheer absence of follow-up imaging.

## 5. Conclusions

In the perspective of the findings, temporalis muscle thickness is a promising surrogate marker for muscle wasting in critically ill neurological patients. All subjects portrayed progressive decline of TMT in ultrasound- as well as CT-based measurements, which was paralleled by muscle wasting of the rectus femoris muscle. Demonstrating good reliability in both TMT measurement techniques, and by evaluating factors potentially confounding the diagnostic accuracy, this study demonstrates the clinical feasibility and utility of temporalis muscle measurement to monitor muscle wasting in a neurocritical care setting.

## Figures and Tables

**Figure 1 nutrients-14-04498-f001:**
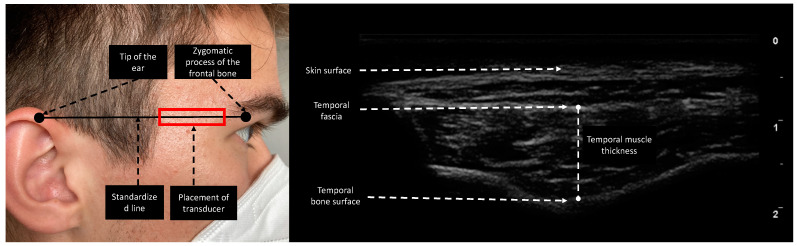
Measurement site of the temporal region. The reference line is the horizontal line connecting the zygomatic process of the frontal bone and the upper edge of the auricula. The transducer was placed rectangular to the skin. TMT was evaluated as the maximal distance between the temporal fascia and temporal bone surface.

**Figure 2 nutrients-14-04498-f002:**
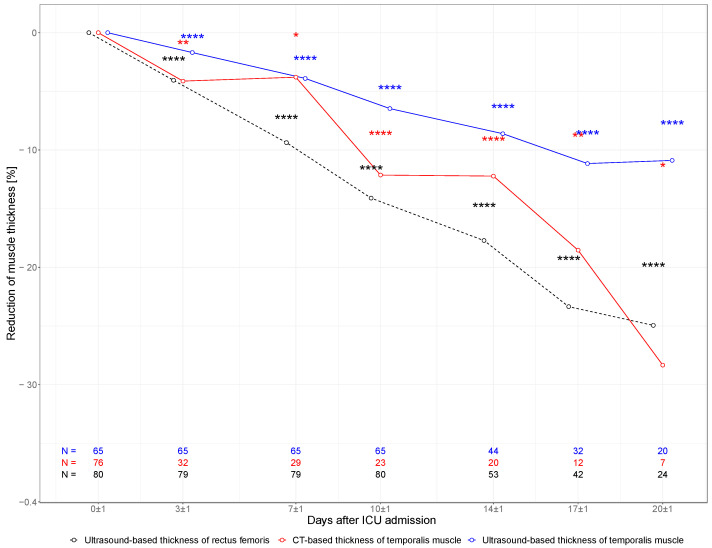
Muscle characteristics. Change in TMT and RFT within the first 20 days of ICU stay. Percent change of muscle thickness in ultrasound-based measurements of TMT (blue line), CT-based TMT (red line), and ultrasound-based RFT (black dotted line). Each measured muscle was treated as a unit of analysis. Correspondingly, n equals double the number of patients included in the study. * *p* < 0.05; ** *p* < 0.01; **** *p* > 0.0001 (*p* values for the Wilcoxon signed-rank test for paired samples referencing day 0).

**Figure 3 nutrients-14-04498-f003:**
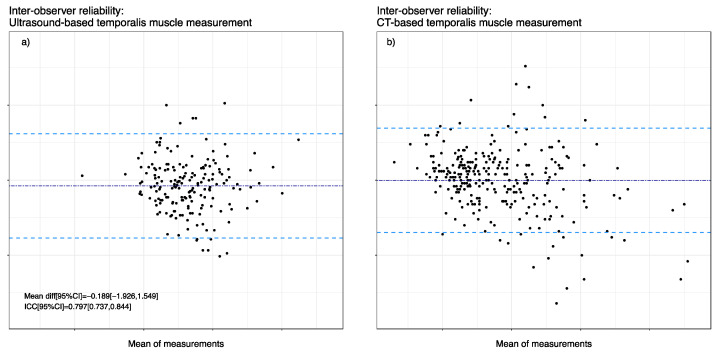
Inter-observer reliability between the two measurement techniques. Bland Altman representation with mean differences (dark blue dashed line) and 1.96 SD limits (light blue dashed lines). (**a**) Inter-observer reliability of ultrasound-based TMT measurements. (**b**) Inter-observer reliability of CT-based TMT measurements.

**Figure 4 nutrients-14-04498-f004:**
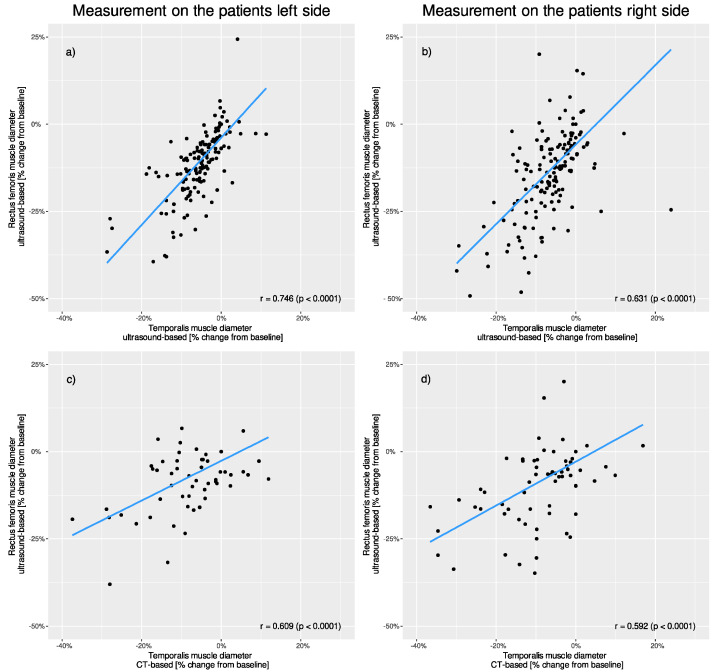
Correlation of percentage changes in muscle thickness between different muscle groups and different measurement techniques. Comparison of the patients’ (**a**) left and (**b**) right-sided ultrasound-based rectus measurements to ultrasound-based temporalis measurements. Comparison of the patients’ (**c**) left and (**d**) right-sided ultrasound-based rectus measurements to CT-based temporalis measurements.

**Figure 5 nutrients-14-04498-f005:**
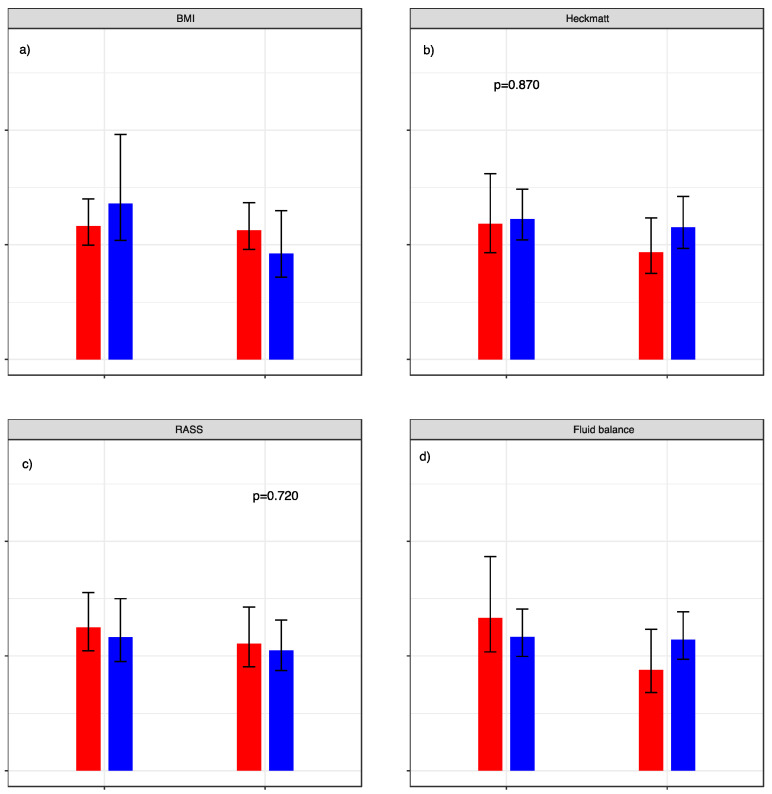
Standard error of measurement relative to patient characteristics. SEM with dichotomization of (**a**) BMI: red graphs represent < 30 versus blue graphs ≥ 30. (**b**) Heckmatt score: red graphs represent cohort with values < 3 and blue graphs ≥ 3. (**c**) RASS score: red graphs represent cohort with values < 0 und blue graphs ≥ 0. (**d**) Fluid retention normalized to body weight: red graphs represent cohort with values < 7.5% (red) versus blue graphs ≥ 7.5%. *p*-values for the calculated F-test are given above the respective graphs.

**Table 1 nutrients-14-04498-t001:** Patient demographics and critical illness parameters.

Parameter	n = 40
Age (years), median [IQR]	60 [52.75–73.5]
Male, n (%)	31 (78%)
BMI (kg/m^2^), mean (SD)	29.3 (4.3)
APACHE II score at ICU admission, median [IQR]	12 [9–18]
NUTRIC score at ICU admission, median [IQR]	3 [2–5]
pMRS = 0 upon admission, n (%)	34 (85%)
Heckmatt score upon admission, median [IQR]	2 [2–3]
ICU admission diagnosis, n (%)	
Intracerebral hemorrhage	12 (20%)
Ischemic stroke	10 (25%)
Subarachnoid hemorrhage	9 (22.5%)
Meningitis/Encephalitis/other neuro-infectious disease	6 (15%)
Status epilepticus	2 (5%)
Cerebral venous sinus thrombosis	1 (2.5%)
Time to baseline ultrasound measurement, days, mean (SD)	0.98 (0.77)
ICU LOS, days, median [IQR]	20 [13–34]
Duration of mechanical ventilation, days, median (IQR)	11.5 [8.5–22.25]
Hospital LOS days, median [IQR]	24 [19–38]
ICU mortality, n (%)	4 (10%)
Nutritional parameters:	
Mean achieved calories from day 1 to 10 in kcal/day (SD)	983.05 (310.12)
Mean achieved protein from day 1 to 10 in g/kg/day (SD)	0.66 (0.42)
Nitrogen balance on day 10 in g/day (SD)	−8.5 (7.1)

IQR = Inter-Quartile Range; SD = Standard Deviation; ICU = Intensive Care Unit; BMI = Body Mass Index; APACHE II = Acute Physiology and Chronic Health Evaluation; NUTRIC = Nutrition Risk in Critically ill score; pMRS = premorbid Modified Rankin Scale; LOS = Length of Stay.

## Data Availability

Data supporting the findings of this study are available from the corresponding author (Konstantinos Dimitriadis) on reasonable request. The data are not publicly available due to ethical restriction.

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
