# Peer review of "Diagnostic Utility of Temporal Muscle Thickness as a Monitoring Tool for Muscle Wasting in Neurocritical Care"

_nutrients, 2022, doi:10.3390/nu14214498_

Round 1

Reviewer 1 Report

The effectiveness of TMT measurement as a method for longitudinally evaluating nutritional status in ICU patients has been verified, and its clinical value is understandable. However, the term 'sarcopenia' appears frequently in the title and text. It is not clear whether the subjects in this study suffer from sarcopenia, and it remains unclear whether TMT measurements are suitable for monitoring sarcopenia. The expression that can be read as saying that TMT can monitor the development of sarcopenia should be revised.

Author Response

Thank you very much for addressing this important aspect. As mentioned by the reviewer, we did not measure direct outcome parameters for evolving sarcopenia such as handgrip strength or volitional functional testing. To address this issue, we rephrased the title and where applicable in the body of the text to “muscle wasting”. Moreover, we added a more detailed description in the introduction in order to disentangle the connection of muscle wasting to ICU acquired weakness and sarcopenia.  

Changes in the manuscript: Where applicable we have adapted the wording according to the recommendation of the reviewer. See changes made in the title, introduction and discussion section.

Reviewer 2 Report

The paper entitled "Diagnostic utility of temporal muscle thickness as a monitoring tool for sarcopenia in neurocritical care" is very interesting the authors demonstrated  the clinical feasibility and utility of ultrasound- and CT 30 based TM measurements for the diagnosis of evolving sarcopenia. The study design is well done and the statistical analysis were convincing. I think this study will be very usefull in clinical activity. 

Author Response

Thank you for very much for your comment.